# Regional Variability of Chestnut (*Castanea sativa*) Tolerance Toward Blight Disease

**DOI:** 10.3390/plants13213060

**Published:** 2024-10-31

**Authors:** Marin Ježić, Lucija Nuskern, Karla Peranić, Maja Popović, Mirna Ćurković-Perica, Ozren Mendaš, Ivan Škegro, Igor Poljak, Antonio Vidaković, Marilena Idžojtić

**Affiliations:** 1Division of Microbiology, Department of Biology, Faculty of Science, University of Zagreb, Marulićev Trg 9a, 10000 Zagreb, Croatia; lucija.nuskern@biol.pmf.hr (L.N.); karla.gregov7@gmail.com (K.P.); mpopovic@sumfak.unizg.hr (M.P.); mirna.curkovic-perica@biol.pmf.hr (M.Ć.-P.); ozren.mendas@gmail.com (O.M.); ivan2.skegro@gmail.com (I.Š.); 2Institute of Forest Engineering, Faculty of Forestry and Wood Technology, University of Zagreb, Svetošimunska 25, 10000 Zagreb, Croatia; 3Institute of Forest Genetics, Dendrology and Botany, Faculty of Forestry and Wood Technology, University of Zagreb, Svetošimunska 23, 10000 Zagreb, Croatia; ipoljak@sumfak.hr (I.P.); avidakovi@sumfak.unizg.hr (A.V.); midzojtic@sumfak.unizg.hr (M.I.)

**Keywords:** disease control, forest ecosystems, phytopathogenic fungi

## Abstract

Since its introduction into Europe in the first half of the 20th century, *Cryphonectria parasitica* has been gradually spreading across the natural range of the sweet chestnut (*Castanea sativa* Mill.), infecting the trees and causing lethal bark cankers. Serendipitously, a hyperparasitic Cryphonectria hypovirus 1 (CHV1), which attenuates *C. parasitica* virulence in combination with more tolerant European chestnut species, was able to ward off the worst effect of the disease. In North America, unfortunately, the native *Castanea dentata* is now functionally extinct since it occurs only as root sprouts in eastern deciduous forests where it was once dominant. In our work, we investigated changes in *C. parasitica* populations over time and the regional variability in chestnut populations’ tolerance toward the blight disease. While vegetative compatibility (vc) type diversity and prevalence of hypovirulence remained similar as in previous studies, in the Buje population, unlike in previous studies, we were unable to find any hypovirulent fungal strains. The most common vegetative compatibility types (vc types) were EU-1, EU-2 and EU-12. However, several rare EU-types were found, including one previously unreported: EU-46. By inoculating several *C. parasitica* strains on tree stems from several chestnut populations, we observed that the induced lesion size was affected by the type of inoculum (CHV1-free or CHV1-infected), genotype-related individual chestnut stem and chestnut stem population of origin-related variability. The largest lesions were induced by CHV1-free fungal isolate DOB-G: 20.13 cm^2^ (95% C.I. 18.10–22.15) and the smallest by CHV1-infected L14/EP713: 2.49 cm^2^ (95% C.I. 1.59–3.39). Surprisingly, the size of the lesions induced by other CHV1-infected strains fell somewhere in between these extremes. The size of induced lesions was dependent on the population of origin as well and ranged from 11.60 cm^2^ (95% C.I. 9.87–13.33) for stems from the Moslavačka gora population to 17.75 cm^2^ (95% C.I. 15.63–19.87) for stems from Ozalj.

## 1. Introduction

Plant pathogens, beyond causing significant damage to their hosts [1], can have an indirect [2,3] or direct impact on humans as well [4] and can cause significant yield loss, making food security, especially in the developing world, problematic [5,6,7]. In forest ecosystems, fungal pathogens can cause significant dieback, disrupting the food webs and microclimatic conditions in forests [8,9], which is especially problematic when alien invasive pathogen species are involved [10]. 

Chestnut blight, caused by the sac fungus *Cryphonectria parasitica* [11], is a disease characterized by cankerous wounds on chestnut bark that cause wilting of distal parts of the infected plants, death of the trees and subsequently, environmental degradation [12]. The fungus is native to eastern Asia, where it causes weak disease symptoms on local chestnut species [13]. After its accidental introduction to North America, it devastated *Castanea dentata* in eastern deciduous forests [12]. In Europe, an RNA virus of the fungus—Cryphonectria hypovirus 1 (CHV1), was introduced with its host, which proved to be serendipitous for the sweet chestnuts (*Castanea sativa* Mill., Fagaceae), as the virus attenuated *C*. *parasitica* virulence toward its tree host. This allowed the chestnuts to recover from the disease and mitigated the effect of chestnut blight in European forests. The virus is now naturally present in most chestnut forests infected with *C*. *parasitica* and is readily utilized in human-mediated biological control of the chestnut blight disease, especially in orchards [14,15]. As the virus lacks an extracellular phase, it can be spread only vertically into conidia or horizontally into another mycelium by hyphal anastomoses [16]. Considering only asexual spores (conidia) carry the virus, sexual reproduction of the host can adversely affect its spread [17]. Horizontal spread is mostly affected by the fungal self/non-self-recognition system, controlled by several vegetative incompatibility (*vic*) loci [18], the combination of which defines specific vegetative compatibility genotypes (vc types). However, as there are signs of migration and sexual reproduction of *C*. *parasitica* in Europe, novel genotypes, i.e., vc types, might be introduced or emerge throughout the range of the fungus, interfering with natural biocontrol [19,20].

In North America, several factors obstructed such a less destructive introductory event scenario: (1) the hypovirulence-inducing virus, CHV1, was not introduced to the continent early during the epidemic, (2) the American chestnut seemed to be more susceptible to the disease and (3) when human-mediated biocontrol measures such as artificial introduction of the CHV1 into American forests were attempted, the virus did not spread efficiently throughout the population [21]. This made biocontrol of the disease in North America especially difficult to achieve. Thus, American scientists’ approach focused more on the selective breeding of resistant chestnuts by crossing highly susceptible American chestnuts with more tolerant Asian species and then back-crossing them, in order to produce tolerant hybrids with as many American chestnut traits as possible [22,23,24]. Similarly, in Europe several hybrid chestnut cultivars were produced by crossing *C. sativa* with *Castanea crenata* or *Castanea mollissima*, resulting in hybrids such as ‘Marigoule’, ‘Bouche de Bétizac’, ‘Marsol’ and ‘Maraval’. However, the practice of growing these cultivars never became widespread throughout the continent [25,26]. An alternative approach was attempted by engineering *C*. *parasitica* super donor strains which allow efficient transmission of CHV1 to various vegetative compatibility types of the fungus [27]. The selection of tolerant genotypes, either by purposefully breeding more tolerant sweet chestnut trees or by crossing susceptible chestnut species with the resistant ones, is one of the solutions to the ever-increasing risk of the spread of various introduced pathogens [24], although the latter received some criticism and setbacks recently.

Therefore, the aims of our work were twofold: (1) resample several populations of *C*. *parasitica* in Croatia to determine the genetic diversity of the fungus and the prevalence of CHV1 and assess the changes in both over the years and (2) detect possible variation in chestnut tolerance toward the disease among several sweet chestnut tree populations utilising biotesting procedures.

## 2. Results

### 2.1. Cryphoncetria parasitica Populations’ Diversity and Cryphonectia hypovirus 1 Prevalence

Thirteen vc types have been identified across four investigated populations among 86 samples. In the populations in Požega and Buje, only five vc types were identified, while in Hrvatska Kostajnica, nine vc types were found. This was reflected in Shannon’s diversity index and evenness (Table 1). In general, the most common vc types were EU-1, EU-2 and EU-12. Overall, none of the identified vc types could be considered dominant in any of the investigated populations. In the population in Cres, we found one new vc type not previously detected in Croatia: EU-46.

Vc type frequency data from previous studies [28,29] were compared with the results obtained in this study: the Kruskall–Wallis test (H 23.46, *p*-value 0.0000497) revealed differences between examined populations, i.e., there were differences in the abundance of certain vc types among all analyzed populations. Thus, the Mann–Whitney pairwise post hoc test was performed between all relevant pairs of analyzed populations. While in most cases the observed differences were not significant, the Buje population experienced changes in frequency and composition of vc types over the years (Table 2). Differences between the populations from Hrvatska Kostajnica sampled in the years 2004 and 2014 were also statistically different, but not the populations obtained in 2004 and 2021, and 2014 and 2021 (Table 2). Interestingly, the populations of Požega and Cres did not show any changes between sampling years 2004 and 2021, although it must be noted they were not sampled and analyzed in 2014.

None of the other comparisons revealed any statistical differences between the composition of the compared populations, i.e., all investigated populations sampled in the same year, whether 2004 or 2021, were similar in composition.

Cryphonectria hypovirus 1 was found in three out of four investigated populations (Table 1); only in Buje, none of the 11 obtained isolates was hypovirulent. In other populations, CHV1 prevalences were as follows: 61% in Hrvatska Kostajnica, 31% in Požega and 50% in Cres. Sequencing of the obtained partial ORFA genomic region revealed that all obtained virus isolates belong to the Italian subtype. The sequences are available in NCBI under accession numbers PP982513–PP982534. A haplonetwork was constructed with the aforementioned partial ORFA sequences and relevant European CHV1 isolates available in NCBI. A highly reticulate network was obtained (Figure 1), indicating no significant differentiation between Croatian CHV1 samples, regardless of the sampling year or population.

### 2.2. Biological Testing

As expected, the inoculations of the CHV1-free *C*. *parasitica* isolates on chestnut stems generally produced the largest lesions, while CHV1-infected strains produced smaller ones. Negative control, i.e., mock inoculation with sterile PDA, did not produce any noticeable lesions beyond small bruising at the place of the wound.

Factorial ANOVA suggested a significant effect of the tree genotype of the individually tested chestnut branches, its population of origin, and the particular inoculum, i.e., combination of a fungal isolate and a particular CHV1 strain, on the lesion size (Table 3). When either only CHV1-free or only CHV1-infected inocula were considered, the effect of the aforementioned categories, i.e., the particular genotype of chestnut branches, the chestnut population of origin and the particular fungal isolate and viral strain combination persisted (Table 3). However, when we considered all fungal inocula (CHV1-free and CHV1-infected) together, inoculum type, i.e., whether a particular inoculum was infected with CHV1 or not, apparently did not contribute significantly to the lesion size (Appendix A). The biogeographical origin of the populations (continental vs. sub-Mediterranean) had only two variables; thus, the differences in lesion sizes obtained on stems from these two groups were only tested with Tukey’s post-hoc HSD and showed statistical significance at *p* = 0.0036.

The lesion sizes ranged from 1.60 cm^2^ (DOB-L inoculum on a stem from Moslavačka gora) to 59.14 cm^2^ (EP155 inoculum on a stem from Buje) for CHV1-free inocula and from 0 cm^2^ (several samples) to 57.53 cm^2^ (L14/Euro 7 inoculum on a stem from Buje) for CHV1-infected inocula. Interestingly, one tree stem from Buje showed particularly large lesion sizes for all inoculated fungal strains (Appendix A). There is a significant overlap in these values and a huge variability depending on a particular inoculum (*C*. *parasitica* vc type and CHV1 viral isolate) and the tree genotype of the inoculated chestnut branch (particular stem and population of origin).

As expected, prototypical hypovirulent (CHV1-infected) isolates, i.e., fungal isolates infected with viruses obtained decades ago, EP713 (F1 subtype) and Euro7 (I subtype), produced smaller lesions than any of the virulent (CHV1-free) isolates. Interestingly, the sizes of the lesions induced by Euro7 were also significantly smaller than the ones induced by EP713 (Figure 2).

The largest overall lesions were induced by CHV1-free isolates with no differences in their size between individual isolates, but lesions induced by two CHV1-infected fungal isolates, L14/CR23 and L14/Euro7, were just as large, indicating that the infection of fungal isolate L14 with viral strains CR23 and Euro7, did not affect fungal virulence, i.e., did not induce a strong hypovirulent effect in the infected mycelium (Figure 2). The smallest, albeit measurable, lesions were induced by another tested CHV1-infected isolate, a combination of L14 fungal isolate and EP713 virus strain (L14/EP713), rather than the original EP713 strain (Figure 2). All this indicates that the infection status of an isolate, i.e., whether it is infected with CHV1 or not, is not necessarily a good predictor of the isolate’s hypovirulence.

The sub-Mediterranean chestnut populations generally had slightly smaller lesion sizes (13.61 cm^2^) than continental populations (14.74 cm^2^) (Figure 3 and Appendix A), with the smallest lesions being found in the population of Poreč (Figure 3). This small difference was nevertheless statistically significant (Tukey HSD *p*-value 0.0037). The largest lesions were observed on stems from sweet chestnuts from the continental population of Ozalj, while the smallest lesions among continental populations were on the stems from Moslavačka gora. Average lesion sizes from other populations fell between these extremes.

Chestnut stems obtained from different individual trees (i.e., unique tree genotypes) showed a huge variability in their reaction to the inoculations. The data considering individual chestnut branch genotypes are given in a Appendix A.

## 3. Discussion

As chestnut blight remains one of the most serious threats to the sweet chestnut in Europe, it is of the utmost importance to follow the disease advancement across the continent and the spread of its biocontrol agent, the virus Cryphonectia hypovirus 1, across the fungal populations, as well as to identify more tolerant chestnut populations, i.e., gene pools, in order to be able to combat the disease on several fronts.

We extended the period of regular monitoring of *C*. *parasitica* populations in Croatia to over 15 years [28,29]. Interestingly, the populations’ composition, i.e., the number of different vc types, their abundance and overall population diversity, did not change considerably over that time. It is worth noting that in some cases we did observe differences, for example, in the sub-Mediterranean population of Buje between samples obtained in 2014 and the years 2004 and 2021 and in the continental population of Hrvatska Kostajnica between samples from 2004 and 2014 [28,29]. However, the 2014 sample size was much larger; thus, several rare vc types were detected in that year, which might have skewed the obtained results [30,31]. On the other hand, sample sizes from the years 2004 and our recent sampling in 2021 are similar in size. That being said, we were able to detect a new vc type for Croatia: EU-46, while vc type EU-26, which was extremely rare in Hrvatska Kostajnica in 2014, is quite common in this study [28]. An observation worth noting was the change in the abundance of vc type EU-12, of which only a single isolate was detected in the population of Cres in 2004, and which is usually associated with southeastern European populations [29,32]. In this study, EU-12 was the most commonly found vc type in the aforementioned population, comprising 35% of the samples. This clonal expansion is known to occur in isolated populations or populations under environmental stress [33,34,35,36,37]. The chestnut population of Cres is isolated on an island and currently in decline with many dead and destroyed trees.

The reason for the relatively high abundance of the CHV1-infected *C*. *parasitica* strains in the population of Cres might be the clonal expansion of local CHV1-infected vc types (e.g., EU-1 and EU-2) as well as the abundance of EU-12, which facilitated the spread of CHV1. The population is not managed, and many of the dead branches are simply left on site. As CHV1 was present among almost all detected *C. parasitica* vc types, this might trigger conidia production, which could be the cause of the spread of these CHV1-infected fungal isolates [15].

In the population of Buje, it is particularly worrying that we have not found a single CHV1-infected (hypovirulent) isolate. The reason for this might be the relatively high number of vc types in that population, which prevents successful natural spread of CHV1. This population always had very low CHV1 prevalence (Krstin et al. 2008; Ježić et al. 2018), which degraded even further. The virus is probably not completely lost from the population but is rather present in an extremely low prevalence, thus making it harder to detect, especially in smaller samples [38].

In biological testing experiments, we inoculated various *C*. *parasitica* strains on chestnut stems from several Croatian chestnut populations. While as expected in any biological system, the variability in response among individual tree genotypes, i.e., inoculated chestnut branches, was quite high, certain patterns did emerge. All CHV1-free strains consistently induced the largest lesions on chestnut stems, with relatively little variability in the lesion sizes, while CHV1-infected strains usually induced smaller lesions, albeit with high variability in their size and occasionally lesions with a size that was comparable with lesions induced by CHV1-free fungal isolates [39,40,41]. For example, the EP713 viral strain in the L14 fungal isolate (L14/EP713) induced overall the smallest lesions on the inoculated stems, thus showing the strongest hypovirulent effect. However, the Euro7 and CR23 viral strains introduced into the same fungal isolate (L14/Euro7 and L14/CR23) induced much larger lesions, larger than the ones induced by the prototypical hypovirulent fungal isolates Euro7 and EP713, and comparable in size to the CHV1-free *C*. *parasitica* isolates. Furthermore, a prototypically stronger EP713 hypovirulent fungal isolate produced slightly larger lesions than the presumably weaker hypovirulent fungal isolate Euro7 [42]. All of this indicates that, as noted in several studies previously, the hypovirulent effect is determined by both the virus and the particular fungal genotype [40,43]. Although it is commonly assumed that CHV1 presence induces a hypovirulent effect, the effect is not uniform across different fungal isolates and viral strains, thus skewing the results when grouping various fungal/CHV1 combinations and positive controls (CHV1-free strains) into the inoculum type category. There is a possibility that this is influenced by a yet undiscovered intrinsic factor, which might affect the hypovirulent effect of Euro7 and EP713 viral strains, similarly as it was shown in *Monilinia fructicola* infected with multiple viruses, altering the hosts’ phenotype [44]. The bottleneck effect on the CHV1 population during the viral transmission from the prototypical strains EP713 or Euro7 into the CHV1-free fungal recipient L14 might have had an effect as well [45].

As stated previously, beyond just the type of inoculum and the particular tree genotype of the inoculated chestnut branch, the lesion size was also affected by the population of origin of the chestnut stems. In other words, some of the studied populations seem to be more tolerant of *C*. *parasitica* infestation while others were shown to be more susceptible. Moreover, in each of the studied regions, sub-Mediterranean and continental, one population stood out as more tolerant: Poreč in the sub-Mediterranean region and Moslavačka gora in the continental region. Interestingly, both these populations are genetically distinct from other studied populations within the respective regions [46,47], and their greater resilience toward chestnut bark might be another example of their uniqueness. In addition, chestnut stems from continental populations showed somewhat lower tolerance, i.e., slightly larger lesion sizes than coastal populations. Continental and sub-Mediterranean chestnut populations in Croatia belong to two different genetic clusters [47]. The genetic cluster of continental populations may be inherently more susceptible to *C*. *parasitica*. The second reason might be the fact that in sub-Mediterranean populations *C*. *parasitica* was established more than half a century ago [48,49], possibly killing off the weaker sweet chestnut genotypes immediately, thus leaving the populations with more resilient genotypes to survive and reproduce [50,51]. This process might still be ongoing in continental populations. Lastly, in continental populations generally more CHV1-free isolates of *C*. *parasitica* can usually be found, thus alleviating the worst of the infection consequences and exerting lower selection pressure on the susceptible sweet chestnut genotypes.

The introduction of the more resistant sweet chestnut genotypes in forests suffering from *C*. *parasitica* infection might provide better long-term outcomes in terms of ecosystem stability and tolerance toward this pathogen.

## 4. Materials and Methods

Chestnut bark samples were collected from four locations in Croatia (Figure 4, Appendix A) in 2021: two continental populations in Hrvatska Kostajnica and Požega and two sub-Mediterranean populations in Buje and Cres. Briefly, from each population, 50 bark cankers (lesions) were sampled, each from only a single tree. Part of the bark, potentially infected with *C*. *parasitica*, was removed and stored in a plastic bag. The knife used for the sampling was sterilized every time a bark sample was removed from a tree by dipping it in 96% ethanol and flaming. The samples were kept at +4 °C until further processing.

From the collected bark samples, pure *C*. *parasitica* cultures (i.e., particular fungal isolates) were obtained by planting small pieces of the bark, the surface of which was sterilized by dipping them for a few seconds in 70% ethanol, drying for a few seconds on a sterile filter paper and inoculating them on ~15 mL PDA (Difco™, BD) in 90 mm Petri dishes. Samples were kept in the dark, at 24 °C and 70% humidity until colony growth was observed as described in [20]. Pure isolates were obtained by transferring hyphal tips to a new Petri dish with fresh PDA. If there was no visible contamination after several days, part of the colony was cut into smaller pieces and frozen in 22% glycerol at −80 °C for long-term storage. The rest of the colony was used for molecular vegetative compatibility typing and determination of CHV1 presence. Thus, a part of the colony was inoculated on a cellophane-covered PDA in order to obtain mycelia for DNA and RNA extractions.

DNA was extracted utilizing OmniPrep for Fungi (G Bioscience) and total RNA was extracted with GenElute™ Total RNA Purification Kit (Sigma Aldrich, Darmstadt, Germany), both following manufacturers’ instructions. The obtained DNA was used for genotyping the isolates on six vegetative compatibility (vc) loci according to [52,53,54]. RNA extracts were used in the RT-PCR protocol in order to detect CHV1 by amplifying part of the ORFA genomic region of the virus according to [55]. The obtained amplicons of the part of the ORFA genomic region of CHV1 were sequenced using a commercially available sequencing service by Macrogen, using the same primer set as in the PCR, thus obtaining reads in both directions. Contigs were assembled using EP721 (DQ861913) as a reference strain with GeneStudio Pro 2.2.

From the obtained genetic data of the *C*. *parasitica* populations, Shannon’s information index, evenness and rarefaction of species richness and Shannon’s information index to the smallest number of isolates (11) were calculated using PAST 4.11 [56] in order to compare the most recent data with previously obtained data. The relevant population groups, i.e., either the same geographical population sampled in different years or all populations sampled within one year, were compared using Kruskall–Wallis and Mann–Whitney pairwise post-hoc tests.

The obtained partial ORFA region CHV1 sequences were aligned and analyzed using Mega 11. Additional sequences used were from Croatian, North Macedonian, Swiss and Montenegrin populations analyzed in previous studies in [20,57], as well as sequences of the representative CHV1 subtypes: AF082191.1 (Euro7) and DQ861913.1 (EP721) of the Italian subtype and NC_001492.1 (EP713) of the French 1 subtype. The obtained alignment was used for haplonetwork construction using a median joining algorithm as implanted in PopART 1.7.

For biotesting of the chestnut susceptibility to *C*. *parasitica*, dormant tree branches approx. 2 cm in diameter and 1 m long were collected from eight populations in Croatia (Figure 4, Appendix A) during the winter season of 2021/2022: four continental populations from Medvednica, Moslavačka gora, Ozalj and Samoborsko gorje and four sub-Mediterranean populations from Buje, Cres, Poreč and Učka. Branches were collected from 8 to 12 trees from each population. Biotesting the susceptibility of the chestnuts was performed according to [40,41]. Several *C. parasitica* isolates were utilized for this: CHV1-free isolates were obtained from populations in Lovran [46]: DOB-G, DOB-I, DOB-L, LD-G and L14, while CHV1-infected isolates were obtained by transferring several CHV1 strains (EP713, Euro7 and CR23) to the L14 *C*. *parasitica* isolate: L14/EP713, L14/Euro7 and L14/CR23 [43,57]. Fungal isolate EP155 was used as a CHV1-free control, while EP713 and Euro7 were used as CHV1-infected controls and a block of sterile PDA was used as negative control, i.e., mock inoculation. Each fungal isolate was inoculated in triplicate on the branches obtained from every tree, i.e., every particular chestnut genotype from a particular population was inoculated three times with each fungal isolate. Holes ~2mm in diameter were drilled in the bark of the chestnut branches and filled with freshly grown mycelia from the aforementioned fungal isolates. The holes were closed with parafilm and inoculated chestnut branches were placed in plastic bags with wet filter paper to keep the moisture high and as constant as possible. The branches were kept at room temperature, occasionally unsealing them to reapply water to the filter papers if necessary and checking the progress of the lesion growth.

Five weeks after the inoculations, the lesions formed on the chestnut branches were measured using callipers: minor and major axes of the elliptical lesions were measured and used to calculate the affected area. The obtained data were analyzed in Statistica 14 (Tibco). Weighted means of the lesions’ area were analyzed by using factorial ANOVA to determine the variables affecting the lesion size: the population of origin of chestnut stems, the particular inoculum used in the experiment, the inoculum type, i.e., CHV1-free, CHV1-infected isolates and negative control (mock inoculation) and the particular genotype of the stem. We compared the lesions’ sizes induced by both CHV1-free and CHV1-infected inocula separately to ascertain the viruses’ effect on the lesion size, i.e., its hypovirulent effect. All inocula (CHV1-free and CHV1-infected) were considered together as well, to control for the inoculum type, i.e., to see which other factors might contribute to the lesion size, regardless of the inoculum. Tukey’s HDS post-hoc test was utilized to determine differences between groups from the aforementioned categories.

## Figures and Tables

**Figure 1 plants-13-03060-f001:**
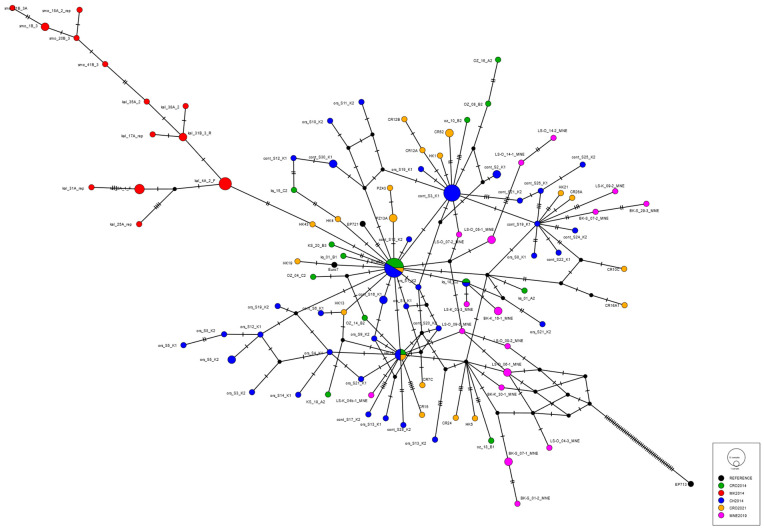
Haplonetwork constructed of partial ORFA region of the Cryphonectria hypovirus 1 (CHV1) sequenced strains using median joining algorithm as implanted in PopART 1.7. Viral strains from different populations are represented with different colors, while the circle size is proportional to the number of detected viral haplotypes (i.e., identical sequence obtained from multiple hypovirulent fungal isolates). Ticks on the lines represent mutations.

**Figure 2 plants-13-03060-f002:**
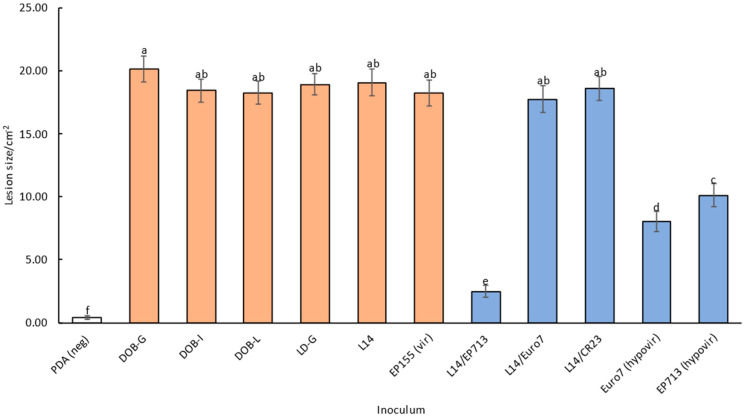
Effect of the inoculum on the lesion sizes. CHV1-free isolates are in orange, CHV1-infected isolates are in blue and mock inoculation is in white. PDA (neg): negative control (mock inoculation); DOB-G, DOB-I, DOB-L, LD-G, L14 and EP155 (vir): CHV1-free inocula; L14/EP713, L14/Euro7, L14/CR23, Euro7 (hypovir) and EP713 (hypovir): CHV1-infected inocula; EP155: CHV1-free inoculum control; Euro7 and EP713: CHV1-infected inoculum control. Significantly different values (Tukey HSD at *p* < 0.05) are indicated with different letters.

**Figure 3 plants-13-03060-f003:**
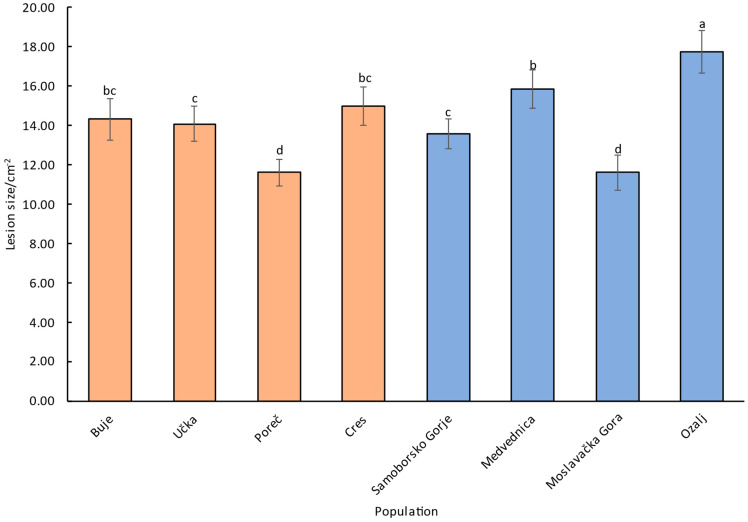
Effect of the stems’ population of origin on the lesion sizes. Sub-Mediterranean populations are represented with orange columns and continental with blue. Significantly different values (Tukey HSD at *p* < 0.05) are indicated with different letters.

**Figure 4 plants-13-03060-f004:**
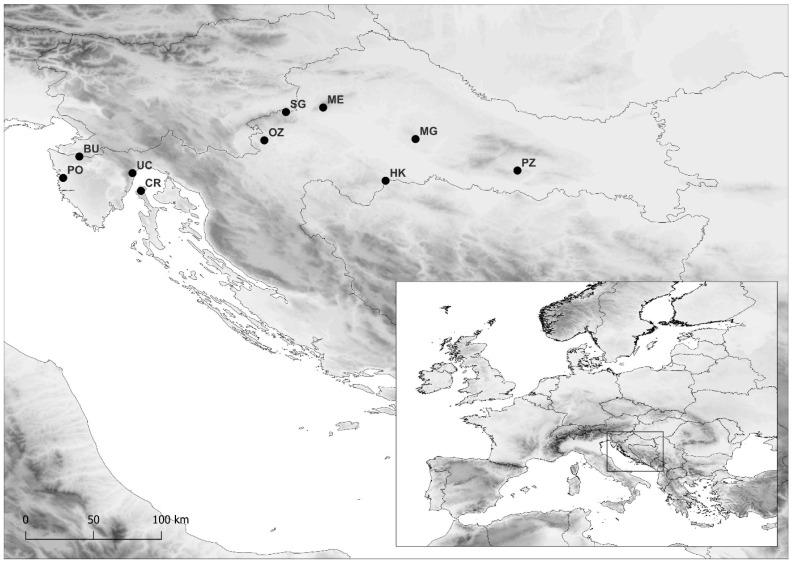
Locations of the sampled sweet chestnut (*Castanea sativa* Mill.) populations. Populations: PO—Poreč; BU—Buje; UC—Učka; CR—Cres; OZ—Ozalj; SG—Samoborsko gorje; ME—Medvednica; HK—Hrvatska Kostajnica; MG—Moslavačka gora, PZ—Požega. Bark cankers were collected in populations HK and PZ and dormant tree branches were collected in PO, UC, OZ, SG, ME and MG, while both types of samples were collected in BU and CR. Geographical coordinates of the populations are given in Appendix A.

**Table 1 plants-13-03060-t001:** Diversity indices of the selected *Cryphonectria parasitica* populations sampled in 2021. A total number of isolates of a specific vc type is given, while their proportion in a population is in parentheses. Shannon’s diversity and evenness are given with a 95% confidence interval.

Vc-Type	Population				
	Hrvatska Kostajnica	Požega	Cres	Buje	Total
EU-1	4 (0.22)	5 (0.19)	4 (0.15)	4 (0.36)	17 (0.20)
EU-2	3 (0.17)	15 (0.58)	2 (0.08)	1 (0.09)	21 (0.24)
EU-4	2 (0.11)	2 (0.08)	-	2 (0.18)	6 (0.07)
EU-5	2 (0.11)	-	-	-	2 (0.02)
EU-6	-	2 (0.08)	-	-	2 (0.02)
EU-12	1 (0.06)	-	9 (0.35)	-	10 (0.12)
EU-13	-	-	5 (0.19)	2 (0.18)	12 (0.14)
EU-17	-	2 (0.08)	-	-	2 (0.02)
EU-18	-	-	1 (0.04)	2 (0.18)	3 (0.03)
EU-21	2 (0.11)	-	-	-	2 (0.02)
EU-24	1 (0.06)	-	-	-	1 (0.01)
EU-26	3 (0.17)	-	-	-	3 (0.03)
EU-46	-	-	5 (0.19)	-	5 (0.06)
Total	18	26	26	11	86
Number of different vc types	9	5	6	5	13
Number of hypovirulent isolates (ratio)	11 (0.61)	8 (0.31)	13 (0.50)	0 (0.0)	55 (0.64)
Shannon diversity index, H’ (95% C.I.)	2.08 (1.74–2.13)	1.23 (0.79–1.44)	1.61 (1.35–1.72)	1.52 (0.99–1.55)	-
Evenness, e (95% C.I.)	0.89 (0.71–0.93)	0.68 (0.51–0.86)	0.84 (0.65–0.94)	0.91 (0.68–0.99)	-
Shannon rarefaction (n = 11)	1.81	1.09	1.45	1.52	-
Richness rarefaction (n = 11)	6.85	3.99	4.92	5	-

**Table 2 plants-13-03060-t002:** Comparison between populations’ composition for populations of *Cryphonectria parasitica* from Buje and Hrvatska Kostajnica between years of sampling. Asterisk indicates populations significantly different in vc types’ distribution.

**Population**	Buje 2004	Buje 2014
Buje 2014	0.0317 *	
Buje 2021	0.5155	0.0030 *
**Population**	Hrvatska Kostajnica 2004	Hrvatska Kostajnica 2014
Hrvatska Kostajnica 2014	0.0135 *	
Hrvatska Kostajnica 2021	0.6622	0.0886

**Table 3 plants-13-03060-t003:** Factorial ANOVA for the effect of a particular genotype of chestnut branches (Genotype), chestnut population of origin (Population), and particular fungal isolate and viral strain combination (Inoculum). Only virus-free fungal isolates are considered in the upper part of the table and only virus-infected fungal isolates are considered in the lower part of the table.

	SS ^a^	Deg. of fr. ^b^	MS ^c^	F ^d^	*p* Value
CHV1-free					
Genotype	308,901.992	67	4610.47749	8.78849266	0 *
Population	13,033.4806	2	6516.74028	12.422211	0.000005 *
Biogeography		0			
Inoculum	394,849.019	6	65,808.1698	125.44354	0 *
CHV1-infected					
Genotype	152,164.671	67	2271.11449	4.01066589	0 *
Population	4902.42698	2	2451.21349	4.32871103	0.0137720962 *
Biogeography		0			
Inoculum	383,493.826	5	76,698.7651	135.445889	0 *

^a^ Sum of squares, ^b^ Degrees of freedom, ^c^ Mean squares, ^d^ F ratio, * statistically significant values (*p* < 0.05).

## Data Availability

Sequence data are publicly accessible in the NCBI database and accession numbers are in the main text. All other data are available as Appendix A.

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
