# Peer review of "Regional Variability of Chestnut (Castanea sativa) Tolerance Toward Blight Disease"

_plants, 2024, doi:10.3390/plants13213060_

Round 1
Reviewer 1 Report
Comments and Suggestions for Authors
1.
1. General Comments
The present work is interesting and contributes to resolving a problem in forest management, namely control of Cryphonectria parasitica spread.
2. Section by section
2.1. - Introduction
This section is very comprehensible, interesting and has recent bibliography to support the discussion made. The authors present a review about the subject explaining in detail the processes involved. For my point of view, a good approach.
2.2. - Material and Methods
Material and Methods are easy to understand and allow to replicate the assay. Statistical methods seems to be appropriated. The schema presented concerning the experimental design is very friendly.
2.3. - Results
Results are well structured with a clear graphic component.
2.4. Discussion
Discussion is well conducted and points out a good possibility to control crops enemies.
3. Suggestion
Extend the study to other sweet chestnut tree populations, namely mediterranean ones.
Author Response
- General Comments
The present work is interesting and contributes to resolving a problem in forest management, namely control of Cryphonectria parasitica spread.
- Section by section
2.1. - Introduction
This section is very comprehensible, interesting and has recent bibliography to support the discussion made. The authors present a review about the subject explaining in detail the processes involved. For my point of view, a good approach.
2.2. - Material and Methods
Material and Methods are easy to understand and allow to replicate the assay. Statistical methods seems to be appropriated. The schema presented concerning the experimental design is very friendly.
2.3. - Results
Results are well structured with a clear graphic component.
2.4. Discussion
Discussion is well conducted and points out a good possibility to control crops enemies.
- Suggestion
Extend the study to other sweet chestnut tree populations, namely mediterranean ones.
Response: We thank the reviewer for the kind words. We concur that extension of the experiments in such a way to include more populations, including more Mediterranean chestnut population would be interesting to investigate. In this research we focused on the continental and Mediterranean sweet chestnut populations from Croatia. According to previous research by Poljak et al. (DOI 10.1007/s11295-017-1107-2), continental Croatian populations form a distinct genetic cluster, different from Mediterranean populations. Furthermore, continental and Mediterranean Croatian populations grow in two distinct climate regions, with continental populations receiving more precipitation, unlike Mediterranean populations which tend to be drier and sunnier. There are differences in soil quality as well, as the Mediterranean populations grow in washed-out karstic soil and are more basic, while continental populations grow on much more acidic substrate. We believe all these differences contribute to potential physiological differences between continental and Mediterranean populations, thus also contributing to the trees’ response to the disease, represented in our work. Surely, it would be interesting to see if similar observations apply for other Mediterranean chestnut populations, unfortunately, this was out of the scope of this research.
Reviewer 2 Report
Comments and Suggestions for Authors
Lesion size, as a metric for the severity of chestnut blight, is affected both by the host genotype and also by the virus. It appears that lesion size is more affected by the latter. In this context, coefficients of determination would be valuable in that that the virus must explain more of the variation in lesion size than does host genotype. Has anyone ever published this finding? It is the responsibility of the authors to address this question, and make it central to their manuscript since the implications apply to plant pathosystems generally, and not just to European chestnut blight.
The first paragraph of the Introduction is largely superfluous and should be deleted. Rewrite the paper so that your central finding [see above] is the thread that you follow!
Comments on the Quality of English LanguageReplace "autochthonous chestnut" with C. dentata in the following, and delete "and Europe" also: "After its accidental introduction to North America and Europe, it devastated autochthonous chestnut forests, altering the local ecosystems [19]".
In this overly long sentence ["Serendipitously, a hyperparasitic Cryphonectria hypovirus 1 (CHV1) which attenuates C. parasitica virulence in combination with more tolerant European chestnut species, was able to ward off the worst effect of the disease, unlike in North America where the native C. dentata has gotten essentially extinct."] C. dentata is now functionally extinct, since it occurs but only as root sprouts; "essentially extinct" is ambiguous.
Author Response
Comments and Suggestions for Authors
Lesion size, as a metric for the severity of chestnut blight, is affected both by the host genotype and also by the virus. It appears that lesion size is more affected by the latter. In this context, coefficients of determination would be valuable in that that the virus must explain more of the variation in lesion size than does host genotype. Has anyone ever published this finding? It is the responsibility of the authors to address this question, and make it central to their manuscript since the implications apply to plant pathosystems generally, and not just to European chestnut blight.
Response 1: We thank the reviewer for the suggestion. Indeed, it is usually assumed that the lesion size is mostly affected by the presence of the virus, which induces the hypovirulent effect in the infected mycelia. In our work, we did not subdivide our fungal strains in virulent (CHV1-free) and hypovirulent (CHV1-infected) categories, as we aimed to control for lesion size induction, i.e., we wanted to see what other factors impact the lesions size, beside the presence or absence of the virus. In our revised manuscript, as suggested by the reviewer, we have expanded our statistical analysis and added additional category to consider: CHV1-free or CHV1-infected, to ascertain the viruses’ effect on the fungus, induction of the hypovirulence and its effect on the lesion size.
These additional analyses are presented in the new Table 3, while the old analysis, where all, CHV1-free and CHV1-infected isolates are grouped together is now presented in Supplementary table 3.
In our work, in fact, the smallest lesions have been induced by L14/EP713 isolate (Figure 2), and not the prototypical hypovirulent isolate EP713, which is considered “strong”, i.e., having the most pronounced hypovirulent phenotype. However, at the same time, other infected fungal strains like L14/Euro7 and L14/CR23 did not show any discernible effect on the lesion size, i.e., there was no discernible hypovirulent effect in these CHV1-infected fungal isolates.
Surprisingly, we were not able to show significant effect of the CHV1 presence/absence in the fungal mycelia on the induced lesion size (Supplementary table 3). While the effect of other considered categories, such as: the population of origin of chestnut stems, the genotype of a particular stem, and the specific combination of the fungal isolate and CHV1 strain, did have generally highly fluctuating (variable) effect on the lesion size, as it is visible from Supplementary table 2, still the effect of a particular category considered was fairly strong and remained statistically significant. The effect of the aforementioned categories on lesion size (i.e., population of origin, the genotype, and the fungal/CHV1 combination) did not vanish even when only CHV1-free or only CHV1-infected fungal isolates were considered alone.
As these additional results did not alter our conclusions, manuscript was moderately expanded. We have added an additional category in our statistical tests, which subdivided our strains for inoculation into virulent or hypovirulent and extended the results (Table 3 and Supplementary table 3), and expanded the main text where appropriate. In order to facilitate the legibility of the manuscript, we have changed in appropriate locations “virulent and hypovirulent” for “CHV1-free and CHV1-infected”, as CHV1-infected and hypovirulent cannot be mutually interchangeable.
The first paragraph of the Introduction is largely superfluous and should be deleted. Rewrite the paper so that your central finding [see above] is the thread that you follow!
Response 2: The first paragraph of the Introduction is essentially a very general introduction to the concept of plant pathogens, their impact on ecosystems and humans. Without it we believe the focus of the Introduction would shift completely. Thus, we have shortened the first paragraph, and left just a few sentences of general introduction for broader audience.
Rephrased paragraph:
Plant pathogens, beyond causing significant damage to their hosts [1], can have an indirect [2,3] or direct impact on humans as well [4] and can cause significant yield loss, making food security, especially in the developing world, problematic [5–7]. In natural ecosystems, like forests, plant pathogenic fungi can cause significant dieback of trees, disrupting the food webs and microclimatic conditions in forests [8,9], which is especially problematic when alien invasive pathogen species are involved [10].
Comments on the Quality of English Language
Replace "autochthonous chestnut" with C. dentata in the following, and delete "and Europe" also: "After its accidental introduction to North America and Europe, it devastated autochthonous chestnut forests, altering the local ecosystems [19]".
Response 3: We thank the reviewer for the suggestion. The sentence has been rephrased:
After its accidental introduction to North America, it devastated C. dentata forests, altering the local ecosystems [19].
In this overly long sentence ["Serendipitously, a hyperparasitic Cryphonectria hypovirus 1 (CHV1) which attenuates C. parasitica virulence in combination with more tolerant European chestnut species, was able to ward off the worst effect of the disease, unlike in North America where the native C. dentata has gotten essentially extinct."] C. dentata is now functionally extinct, since it occurs but only as root sprouts; "essentially extinct" is ambiguous.
Response 4: We thank the reviewer for the suggestion. The sentence has been rephrased:
Serendipitously, a hyperparasitic Cryphonectria hypovirus 1 (CHV1) which attenuates C. parasitica virulence in combination with more tolerant European chestnut species, was able to ward off the worst effect of the disease. In North America, unfortunately, the native C. dentata is now functionally extinct since it occurs only as root sprouts.
Reviewer 3 Report
Comments and Suggestions for Authors
The article focuses on the regional variability ( Croatia)of the tolerance of chestnut (Castanea sativa) against the canker disease, caused by the fungus Cryphonectria parasitica. Since its introduction to Europe in the 20th century, the fungus has devastated chestnut forests. However, the presence of the hypovirulent Cryphonectria hypovirus 1 (CHV1) has mitigated the severity of the disease in Europe, allowing European chestnut trees to partially recover, unlike in North America, where C. dentata has almost disappeared.
The study evaluated changes in C. parasitica populations and variability in chestnut tolerance over time in different regions of Croatia. It was observed that the lesions caused by the fungus varied depending on the type of inoculation, the genotype of the chestnut trees and the region of origin of the trees. The virulent strains caused larger lesions, while the hypovirulent strains, although they mostly generated smaller lesions, showed great variability. Regarding C. parasitica populations, some regions showed a significant prevalence of hypovirulent strains, indicating active biological control of the disease, while in others, such as Buje, no hypovirulent strains were found, which could make it difficult to control the disease. natural control of the disease.
The authors conclude that some populations of chestnut trees appear to be more tolerant to the disease, especially in regions where C. parasitica has been present for longer, suggesting a process of natural selection. In addition, they highlight the importance of continuing to monitor the disease and explore biological control strategies to mitigate its impact on ecosystems.
The introduction is sufficient and relevant. Materials and Methods are clearly defined. The presentation of the results (both in the text and in the figures) is also clear. I recommend acceptance in the present form.
Author Response
The article focuses on the regional variability ( Croatia)of the tolerance of chestnut (Castanea sativa) against the canker disease, caused by the fungus Cryphonectria parasitica. Since its introduction to Europe in the 20th century, the fungus has devastated chestnut forests. However, the presence of the hypovirulent Cryphonectria hypovirus 1 (CHV1) has mitigated the severity of the disease in Europe, allowing European chestnut trees to partially recover, unlike in North America, where C. dentata has almost disappeared.
The study evaluated changes in C. parasitica populations and variability in chestnut tolerance over time in different regions of Croatia. It was observed that the lesions caused by the fungus varied depending on the type of inoculation, the genotype of the chestnut trees and the region of origin of the trees. The virulent strains caused larger lesions, while the hypovirulent strains, although they mostly generated smaller lesions, showed great variability. Regarding C. parasitica populations, some regions showed a significant prevalence of hypovirulent strains, indicating active biological control of the disease, while in others, such as Buje, no hypovirulent strains were found, which could make it difficult to control the disease. natural control of the disease.
The authors conclude that some populations of chestnut trees appear to be more tolerant to the disease, especially in regions where C. parasitica has been present for longer, suggesting a process of natural selection. In addition, they highlight the importance of continuing to monitor the disease and explore biological control strategies to mitigate its impact on ecosystems.
The introduction is sufficient and relevant. Materials and Methods are clearly defined. The presentation of the results (both in the text and in the figures) is also clear. I recommend acceptance in the present form.
Response: We graciously thank the reviewer for the kind words.
Round 2
Reviewer 2 Report
Comments and Suggestions for Authors
"unfortunately, the native C. dentata is now functionally extinct since it occurs only as root sprouts". Correct to: unfortunately, the native C. dentata is now functionally extinct since it occurs only as root sprouts in eastern deciduous forests where it was once dominant.
"After its accidental introduction to North America, it devastated C. dentata forests, altering the local ecosystems [14]." Correct to: After its accidental introduction to North America, it devastated C. dentata in eastern deciduous forests.
". In natural ecosystems, like forests, plant pathogenic fungi can cause significant dieback of tree". Correct to: In forest ecosystems fungal pathogens can cause significant dieback.
Make sure that you consistently distinguish tree genotypes from fungal genotypes. In places you just mention 'genotypes' without saying that they were tree or fungal.
Shannon was mispelled at least once.
Author Response
Comments 1: "unfortunately, the native C. dentata is now functionally extinct since it occurs only as root sprouts". Correct to: unfortunately, the native C. dentata is now functionally extinct since it occurs only as root sprouts in eastern deciduous forests where it was once dominant.
Response1: We thank the reviewer for the suggestion and have corrected the text per their suggestion.
Comments 2: "After its accidental introduction to North America, it devastated C. dentata forests, altering the local ecosystems [14]." Correct to: After its accidental introduction to North America, it devastated C. dentata in eastern deciduous forests.
Response 2: We thank the reviewer for the suggestion and have corrected the text per their suggestion.
Comments 3: ". In natural ecosystems, like forests, plant pathogenic fungi can cause significant dieback of tree". Correct to: In forest ecosystems fungal pathogens can cause significant dieback.
Response 3: We thank the reviewer for the suggestion and have corrected the text per their suggestion.
Comments 4: Make sure that you consistently distinguish tree genotypes from fungal genotypes. In places you just mention 'genotypes' without saying that they were tree or fungal.
Response 4: We thank the reviewer for the suggestion to add additional explanation when distinguishing tree genotypes form fungal genotypes. On several places in text, we have added in the appropriate places either “tree genotype” or “sweet chestnut genotype” for additional clarity.
Comments 5: Shannon was mispelled at least once.
Response 5: We thank the reviewer for noticing and went over the text using “find” option. We were able to find one more instance of misspelled “Shannon” in Table 1 and have corrected it.